# Safety and Immunogenicity of the Tetravalent Recombinant COVID-19 Protein Vaccine SCTV01E in Children and Adolescents Aged 3 to 17 Years: A Randomized, Double-Blind, Placebo-Controlled Phase 2 Clinical Trial

**DOI:** 10.3390/vaccines13010043

**Published:** 2025-01-07

**Authors:** Fengcai Zhu, Ting Huang, Pengfei Jin, Linglin Zhang, Zhongqiang Jin, Wenli Zhang, Dongya Yuan, Zhong Wang, Yusong Deng, Jiaxin Li, Xiao Shen, Yongpan Fu, Jian Li, Xinjie Yang, Jing Li, Liangzhi Xie

**Affiliations:** 1Jiangsu Provincial Center for Disease Control and Prevention, Jiangsu Provincial Academy of Preventive Medicine, Nanjing 210009, China; jszfc@vip.sina.com (F.Z.); jpf19891103@163.com (P.J.); 2Sichuan Center for Disease Prevention and Control, Chengdu 610041, China; cocoht@163.com (T.H.); xiaoshuanxiaozhu@163.com (L.Z.); zhongqiang_jin@163.com (Z.J.); wenli_zhang2024@163.com (W.Z.); 3Dazhu County Disease Prevention Control Center, Dazhu 635199, China; dongya_yuan11@163.com (D.Y.); zhong_wang12@163.com (Z.W.); yusong_deng@163.com (Y.D.); jiaxin_li11@163.com (J.L.); xiao_shen11@163.com (X.S.); 4Beijing Engineering Research Center of Protein and Antibody, Sinocelltech Ltd., Beijing 100176, China; yongpan_fu@sinocelltech.com (Y.F.); jian_li@sinocelltech.com (J.L.); xinjie_yang@sinocelltech.com (X.Y.); jing_li5@sinocelltech.com (J.L.); 5Cell Culture Engineering Center, Chinese Academy of Medical Sciences & Peking Union Medical College, Beijing 100006, China

**Keywords:** children and adolescents, immunogenicity, safety, multivalent vaccine, SARS-CoV-2

## Abstract

**Background**: SCTV01E is a tetravalent recombinant COVID-19 vaccine authorized for emergency use in China for adults 18 years and older but not for those under 18. **Objective**: This Phase 2 trial assessed the safety and immunogenicity of SCTV01E in healthy children and adolescents aged 3 to 17 years, to establish immunobridging with that observed in adults from the efficacy pivotal trial (NCT05308576). **Methods**: Participants were randomly assigned to receive either 30 µg of SCTV01E or a placebo. Primary endpoints were safety and immunogenicity focused on the geometric mean titer (GMT) and seroresponse rate (SRR) of neutralizing antibodies (nAb) against Omicron BA.5. **Results**: In total, 268 participants (214 SCTV01E vs. 54 placebo) were included in the safety analysis, with 241 participants (191 vs. 50) in the immunogenicity analysis. Overall, 127 (59.3%) participants receiving SCTV01E and 9 (16.7%) receiving a placebo reported adverse events (AEs), most of which were Grade 1 or 2. No serious adverse events (SAEs) or adverse events of special interest (AESIs) were reported. In the immunogenicity bridging analysis, data from 95 youths were compared with data from 188 adults; the geometric mean ratio (GMR) of the titers was 8.78 (95% CI: 6.05–12.74, *p* < 0.001), with the lower bound of the 95% CI exceeding 0.67. The difference in the SRR was 6.34% (95% CI: 0.93–11.22%) (*p* = 0.029), and the lower bound of the 95%CI was >−5%, indicating superiority. **Conclusions**: SCTV01E was found to be safe and well tolerated in children and adolescents, generating a robust immune response against Omicron BA.5. This supports its potential use in younger populations.

## 1. Introduction

After first being detected in 2019, the outbreak of COVID-19, caused by severe acute respiratory syndrome coronavirus-2 (SARS-CoV-2), became a global pandemic and besieged the world for more than four years, resulting in more than 7 million deaths [1,2]. With the deployment of effective pandemic management, and the widespread use of COVID-19 vaccines against COVID-19, the life-threatening disease is now under control. Although the World Health Organization (WHO) declared the end of the pandemic, COVID-19 disease is still ongoing [3]. SARS-CoV-2 Omicron subvariants KP.2, KP.2.3, KP.3, KP.3.1.1, and LB.1 have a high prevalence, and KP.3.1.1 accounts for approximately 37% of new COVID-19 cases in the USA [4]. Although vaccines based on earlier variants are still effective in reducing severe disease [5], multivalent vaccines with cross-reactivity may address the issue of virus mutation and provide additional protection.

Based on the first-generation COVID-19 protein vaccine (SCTV01C, which was designed to target the SARS-CoV-2 Alpha and Beta variants and received Emergency Use Authorization on 4 December 2022), SCTV01E, a tetravalent recombinant COVID-19 protein vaccine, was developed against SARS-CoV-2 Alpha, Beta, Delta, and Omicron BA.1 variants. In a pivotal Phase 3 clinical trial conducted in China (NCT05308576) with 9196 adult participants, SCTV01E was proved to be safe and effective in protecting people from symptomatic infection, with a vaccine efficacy of 69.4% (95% CI: 50.6, 81.0) [6]. The data supported the authorization of the emergency use of SCTV01E in adults aged 18 years and older in the context of a booster vaccination on 22 March 2023. However, the safety and immunogenicity of SCTV01E in children and adolescents aged 3 to 17 years is still unknown. To investigate the potential use of SCTV01E in children and adolescents, we conducted a randomized, double-blind, placebo-controlled Phase 2 clinical trial. Immunogenicity data were compared with those from the immunogenicity subgroup of the Phase 3 trial in adults aged 18 years and older.

## 2. Materials and Methods

### 2.1. Trial Design

This randomized, double-blind, placebo-controlled Phase 2 clinical trial evaluated the safety and immunogenicity of SCTV01E in healthy children and adolescents aged 3 to 17 years. This trial was conducted at the Dazhu Center for Disease Control and Prevention (Dazhu, China). The trial followed the ethical requirements of Good Clinical Practice and the Declaration of Helsinki. The National Medical Products Administration (NMPA) of China approved SCTV01E as an investigational new drug (IND) (No.: 2022L90019). Additionally, the Ethics Committee of the Sichuan Provincial Center for Disease Control and Prevention approved the protocol, informed consent, and study amendments (Ethics number: SC-0820224301).

### 2.2. Study Population

Participants aged 3 to 17 years were screened for eligibility. Key inclusion criteria included previous COVID-19 vaccination (primary or booster), the time interval between informed consent and study injection (6 to 24 months), and healthy or stable health conditions. Exclusion criteria included those with diagnosed, confirmed, or suspected SARS-CoV-2 infection; and axillary temperature ≥ 37.3 °C within 72 h prior to study vaccination. Before recruitment, legal guardians for children aged 3 to 7, participants aged 8 to 17 and their legal guardians provided written informed consent. Data from the immunogenicity subgroup (n = 188) of a randomized, double-blind, placebo-controlled trial (NCT05308576) were analyzed to perform immunogenicity bridging between adults and participants aged 3 to 17 years.

### 2.3. Randomization and Masking

An independent statistician from a third party generated randomization codes by block randomization using SAS software (version 9.4). Eligible participants were randomized to receive one dose of 30 µg SCTV01E (0.5 mL) or placebo (normal saline 0.5 mL) in a ratio of 4:1 using the Interactive Network Response System (IWRS). Randomization stratification factors for participants aged 12 to 17 years were previous COVID-19 vaccination (primary vs. booster), type of COVID-19 vaccine last received (inactivated vs. non-inactivated), and the time interval between informed consent and last vaccination (6–12 months vs. 12 to 24 months). For participants aged 3 to 11 years, the randomization stratification factors were age (3 to 5 years vs. 6 to 11 years), previous COVID-19 vaccination (primary vs. booster), type of COVID-19 vaccine last received (inactivated vs. non-inactivated), and the time interval between informed consent and last vaccination (6–12 months vs. 12–24 months). All participants, investigators, clinical research associates, data analysts, and laboratory staff were blind to group allocation.

### 2.4. Study Interventions

SCTV01E is a tetravalent COVID-19 protein vaccine, composed of the trimeric spike extracellular domain (S-ECD) from four SARS-CoV-2 variants, specifically Alpha, Beta, Delta, and Omicron BA.1, with a squalene-based oil-in-water adjuvant SCT-VA02B. It is derived from the earlier bivalent vaccine, SCTV01C, which targeted the Alpha and Beta variants. In response to the emergence of new variants and the need to address the ongoing pandemic, a series of products have been created on the same platform. These have been subjected to rigorous evaluation in clinical trials conducted in both domestic and international settings, with a focus on assessing the safety, immunogenicity, and VE of the vaccines [6,7,8,9,10,11,12].

### 2.5. Study Procedures

As this was the first time that SCTV01E was administered to children and adolescents aged 3 to 17 years, 15 sentinel participants aged 12 to 17 years were initially enrolled for safety reasons and monitored by the Data and Safety Monitoring Board (DSMB) for 7 days after vaccination. Once there were no safety concerns in the sentinel participants, the study proceeded with the enrollment of the remaining participants aged 12 to 17 years and sentinel participants aged 3 to 11 years. This included 15 sentinel participants aged 6 to 11 years and 15 aged 3 to 5 years. Study vaccines were administered intramuscularly into the upper arm of each participant.

### 2.6. Safety and Immunogenicity Assessments

For safety assessment, each participant remained at the vaccination center for at least 30 min for observation. Predefined solicited AEs (local and systemic) were collected within 7 days after injection on a Vaccination Record Card (VRC), while non-predefined unsolicited AEs were collected within 28 days after injection. SAEs and AESIs were collected throughout the trial (usually within 180 days). We used NMPA toxicity criteria in preventive vaccine clinical trials to grade AEs.

Eligible participants provided blood samples before vaccination, and 28, 90, and 180 days after injection to assess neutralization of SARS-CoV-2 Omicron BA.5 (PRNT50). Total anti-SARS-CoV-2 IgG was assessed only before vaccination (ELISA).

### 2.7. Study Outcomes

The primary endpoints were incidence and severity of solicited AEs, geometric mean titer (GMT), and seroresponse rate (SRR) of nAb against Omicron BA.5 28 days after vaccination and immunogenicity bridging assessment based on the GMR and SRR between 3 to 17 years old participants and adult participants from the pivotal Phase 3 trial. The secondary endpoints were the incidence and severity of unsolicited AEs, SAEs, AESIs, GMT, and SRR of nAb against Omicron BA.5 90 and 180 days after vaccination. The SRR of nAb was defined as a change from a value below the low limit of quantitation (LLOQ) to a value equal to or above the LLOQ or a ≥4-fold rise if the baseline is equal to or above the LLOQ in nAb to Omicron BA.5.

### 2.8. Statistical Analysis

The safety population included all participants who received either SCTV01E or a placebo. Safety results were presented as counts, percentages, and two-sided 95% confidence intervals (CI) using the Clopper–Pearson method. AEs, SAEs, and adverse drug reactions (ADRs) were categorized according to the Medical Dictionary for Regulatory Activities (version 26.1).

An immunogenicity assessment was conducted in participants aged 3 to 17, as well as in age-specific cohorts: 12 to 17, 6 to 11, and 3 to 5. The immunogenicity population included those who were randomized, received the injection per protocol, and had valid blood test results before and after vaccination, meeting all eligibility criteria. To establish immunogenicity bridging between adults and youths, adults with a baseline anti-spike IgG level < 338 BAU/mL (an immunogenicity subgroup from the Phase 3 trial) were compared with youths aged 3 to 17 years who also had a baseline IgG level < 338 BAU/mL. The threshold of 338 BAU/mL was chosen to minimize the potential interference of prior infections on vaccine-induced immune responses. The sample size was determined using a non-inferiority design, aiming to demonstrate that the immune response to SCTV01E in participants aged 3 to 17 years was non-inferior to that in adults, as measured by the GMR and SRR of neutralizing antibodies (nAbs) against Omicron BA.5. Approximately 107 participants aged 3 to 17 years were estimated to provide 80% power to demonstrate non-inferiority in the GMR, defined as the lower bound of the 95% CI exceeding 0.67. Similarly, a sample size of approximately 105 participants was estimated to provide 80% power to demonstrate non-inferiority in the SRR, defined as the lower bound of the 95% CI for the difference in the SRR being greater than −5%.

## 3. Results

### 3.1. Participants, Demographics, and Baseline Characteristics

Between 5 January and 11 December 2023, 316 children and adolescents were screened for eligibility, and 268 eligible participants were randomized and received SCTV01E or a placebo (12–17 years, n = 79; 6–11 years, n = 140; 3–5 years, n = 49). In the 12–17 years age group, 63 received SCTV01E while 16 received placebo. In the 6–11 years age group, 112 received SCTV01E and 28 received a placebo. In the 3–5 years age group, 39 received SCTV01E and 10 received a placebo (Figure 1). The median age of the children and adolescent participants was 8 years (range: 3–17 years), while the median age of adults from the Phase 3 trial for immunobridging was 53.5 years (range: 18–82 years). The proportion of male participants was 54.2% in the youth cohort and 67.6% in the adult cohort. All children and adolescent participants (100%) had previously received two doses of the primary vaccination with the inactivated vaccine. In contrast, among adult participants, 94.7% had received a booster dose, with 64% receiving inactivated vaccines and 35.6% receiving non-inactivated vaccines. The interval between the last vaccination and the administration of the study vaccine was 6–12 months for 2.8% of participants aged 3–17 years and 42.0% of participants aged ≥18 years. The interval was 12–24 months for 97.2% of participants aged 3–17 years and 58.0% of participants aged ≥18 years. Other demographic characteristics and baseline information are provided in Table 1.

### 3.2. Safety Profiles

The prevalence of treatment-related adverse events (TRAEs) in participants aged 3 to 17 in the Phase 2 study was 51.4% in the SCTV01E group and 13.0% in the placebo group. A greater proportion of children and adolescent participants who received SCTV01E reported local and systemic AEs than those who received a placebo (Figure 2). The local and systemic AEs were mostly mild to moderate in severity. Overall, 56 (26.2%) participants aged 3 to 17 years in the SCTV01E group reported local AEs, compared to 1 (1.9%) in the placebo group. The common local AEs were injection site pain (SCTV01E vs. placebo, 22.9% vs. 1.9%), swelling (7.9% vs. 0%), erythema (5.6% vs. 0%), induration (3.3% vs. 0%), and pruritus (1.9% vs. 0%); all of them were Grade 1 and 2, and recovered without any intervention. Local AE rates were similar among groups aged 12–17 years (25.4%), 6–11 years (30.4%), and 3–5 years (15.4%). The common systemic AEs were fever (SCTV01E vs. placebo, 32.2% vs. 1.9%), headache (13.6% vs. 1.9%), fatigue (12.1% vs. 0%), cough (4.2% vs. 0%), vomiting (4.2% vs. 0%), chills (3.3% vs. 0%), muscle pain (2.8% vs. 1.9%), nausea (2.8% vs. 1.9%), and joint pain (1.9% vs. 0%). There were five (2.3%) children and adolescents who reported Grade 3 fever related to SCTV01E. No other Grade 3 and above AE related to SCTV01E was reported. No SAE or AESI was reported throughout the trial.

### 3.3. Immunogenicity Outcomes

A total of 191 children and adolescents who received SCTV01E were included in the immunogenicity assessment. The baseline nAb GMT against Omicron BA.5 was 54 (95% CI: 40–74). On Day 28, the antibody titer increased 28.9-fold, reaching a value of 1564 (1378–1774). On Days 90 and 180, the fold change remained at 22.3 (14.58–34.07) and 15.6 (9.98–24.39), and the GMTs were 1364 (1187–1567) and 954 (812–1121), respectively. In the age group of 12 to 17 years, the baseline nAb GMT against Omicron BA.5 was 67 (37, 119), on Day 28, it increased 19.3 (10.54, 35.36)-fold to 1314 (1002, 1724), 17.6 (7.58, 40.74)-fold and 11.9 (4.52, 31.44)-fold on Day 90 and 180, to 1146 (820, 1601) and 1149 (721, 1833), respectively. In the age group of 6 to 11 years, the nAb GMT against Omicron BA.5 increase was 38.5 (25.18–58.99)-fold on Day 28, 28.2 (15.23–52.35)-fold on Day 90, and 22.8 (12.56–41.51)-fold on Day 180. In the age group of 3 to 5 years, the nAb GMT against Omicron BA.5 increase was 22.8 (12.15–42.95)-fold on Day 28, 17.7 (7.35–42.83)-fold and 6.7 (2.62–17.05)-folds on Day 90 and 180. The results indicated that SCTV01E induced a robust and sustained immune response in children and adolescents aged 3 to 17 years (Figure 3).

An immunogenicity subgroup from the randomized, placebo-controlled trial (NCT05308576), consisting of 188 adults aged over 18 years and 95 youths aged 3 to 17 years, was compared to establish immunogenicity bridging between the two groups. Twenty-eight days after vaccination, the GMT of nAb against Omicron BA.5 increased 162-fold from the baseline value of 9 (95% CI: 7–11) to 1290 (1077–1545) in participants aged 3 to 17 years. In adults, the increase was 28.6-fold from baseline 4 (4–5) to 123 (98–155). The GMR was 8.78 (95% CI: 6.05–12.74) (*p* < 0.001), with a lower bound of the 95% CI greater than 0.67. The SRR of neutralizing nAb against the Omicron BA.5 variants was 98.9% (95% CI: 93.96–99.97%) in participants aged 3 to 17 years. In participants aged 18 years and older, the SRR was 92.6% (87.82–95.87%), with an SRR difference of 6.34% (95% CI: 0.93–11.22%). The lower bound of the 95% CI was greater than −5%. The GMR and SRR results met the predefined criteria for a non-inferior endpoint, further superiority (Table 2).

## 4. Discussion

SCTV01E is applied in immunization as a booster dose in the population aged 18 years and older in China according to Emergency Use Authorization, and the Biologics License Application process is ongoing. The favorable safety profile and robust immune responses of SCTV01E observed in adults prompted an evaluation of the safety and immunogenicity of SCTV01E in children and adolescents. This study aims to bridge the immune response gap between these age groups and adults. The pandemic has significantly affected children and adolescents, particularly those living with elderly family members who have underlying health conditions. Although children are not the primary drivers of SARS-CoV-2 transmission, they can still spread the virus to vulnerable family members, a scenario commonly observed in Chinese communities.

The incidence of TRAE in participants aged 18 years and older in the Phase 3 trial was 16.8% and 6.2% in the SCTV01E and placebo groups, respectively [6]. In this Phase 2 trial, 51.4% of participants aged 3 to 17 in the SCTV01E group reported TRAEs, compared to 13.0% in the placebo group. TRAEs were more common in this age group than in adults, but most were mild to moderate in severity. The incidences of TRAEs in age groups of 12 to 17 years (42.9%), 6 to 11 years (58.0%), and 3 to 5 years (53.8%) were similar. The incidence of TEAEs for the 3–17 age cohort in our trial was 59.3%. Contrastingly, TEAEs for another approved recombinant COVID-19 vaccine (ZF2001) in children and adolescents aged 3 to 17 was 45%, with 22% related to the vaccine used during primary vaccination. This is similar to the approximately 40% TEAEs observed in adults aged 18 and older [13]. Recombinant COVID-19 protein vaccines generally have fewer adverse reactions than mRNA COVID-19 vaccines in children and adolescents, so in mRNA vaccine clinical trials involving participants aged 6 months to 17 years, the doses were down-regulated for safety reasons [14,15,16,17,18,19,20]. The safety profile suggests that SCTV01E is safe and tolerable and supports its application in children and adolescents.

Neutralizing antibody concentrations is considered a surrogate index of vaccine efficacy [21]. We previously conducted a Phase 3 clinical trial to evaluate the vaccine efficacy and safety of SCTV01E in participants aged 18 years and older who received primary or booster COVID-19 vaccines between 26 December 2022 and 15 January 2023, and demonstrated for the first time effectively preventing symptomatic and asymptomatic COVID-19 infection, during a period when Omicron BA.5 was the dominant variant. In this study, we compared the GMTs and SRRs of young participants aged 3 to 17 with those of adult participants aged 18 and older in the Phase 3 trial. All participants included in the immunobridging analysis had a baseline total anti-spike IgG level below 338 BAU/mL. The threshold of 338 BAU/mL was chosen to minimize the potential interference of prior infections on vaccine-induced immune responses. This cutoff value was determined based on findings from previous immunogenicity trials involving protein-based vaccines (NCT05043285, NCT05043311, NCT05323461). These studies identified a significant correlation between anti-spike IgG levels and the risk of SARS-CoV-2 infection. Specifically, when IgG levels exceed 338 BAU/mL, it indicates a high likelihood of a recent infection, which is approximately 10 times the lower limit of detection of the assay used in those studies.

SCTV01E induced robust neutralizing antibody responses against the Omicron BA.5 variant in 3 to 17 and ≥18 years old participants. The immunobridging analysis proved that SCTV01E induced immunogenicity in the 3 to 17 year group, which was non-inferior to that observed in participants aged 18 and older according to more stringent criterion in the SRR difference established by the NMPA compared to FDA recommendations (greater than −5% vs. −10%) [15].

The trial has several limitations. Firstly, the primary focus of this study was on safety and immunogenicity bridging; however, vaccine efficacy (VE) was not assessed. Secondly, the immunobridging analysis was performed based on the adult population, which included all participants aged 18 years and older, and the youth population, which comprised those aged 3 to 17 years. Detailed analysis of the sub-age categories was not performed due to the limited sample size. Nevertheless, related immunobridging studies of youths and adults have indicated that the trends observed in the sub-age groups were similar to those seen in the overall population [13,14,15,16,17,18,19,20].

## 5. Conclusions

The trial results demonstrate that SCTV01E is safe, well tolerated, and immunogenic for children and adolescents aged 3 to 17 years, showing non-inferiority to adults. The findings support the application and further study of SCTV01E in children and adolescents.

## Figures and Tables

**Figure 1 vaccines-13-00043-f001:**
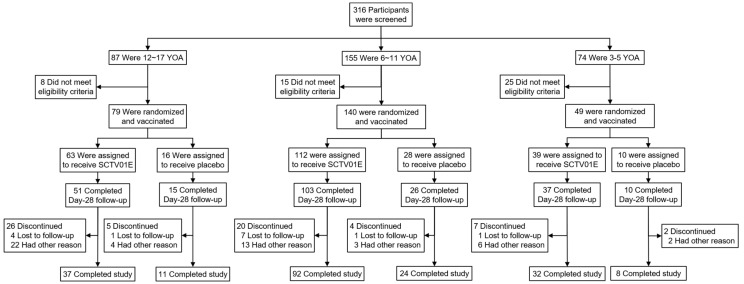
Screening, randomization, and vaccine and placebo administration among participants at 3 to 17 years of age (YOA).

**Figure 2 vaccines-13-00043-f002:**
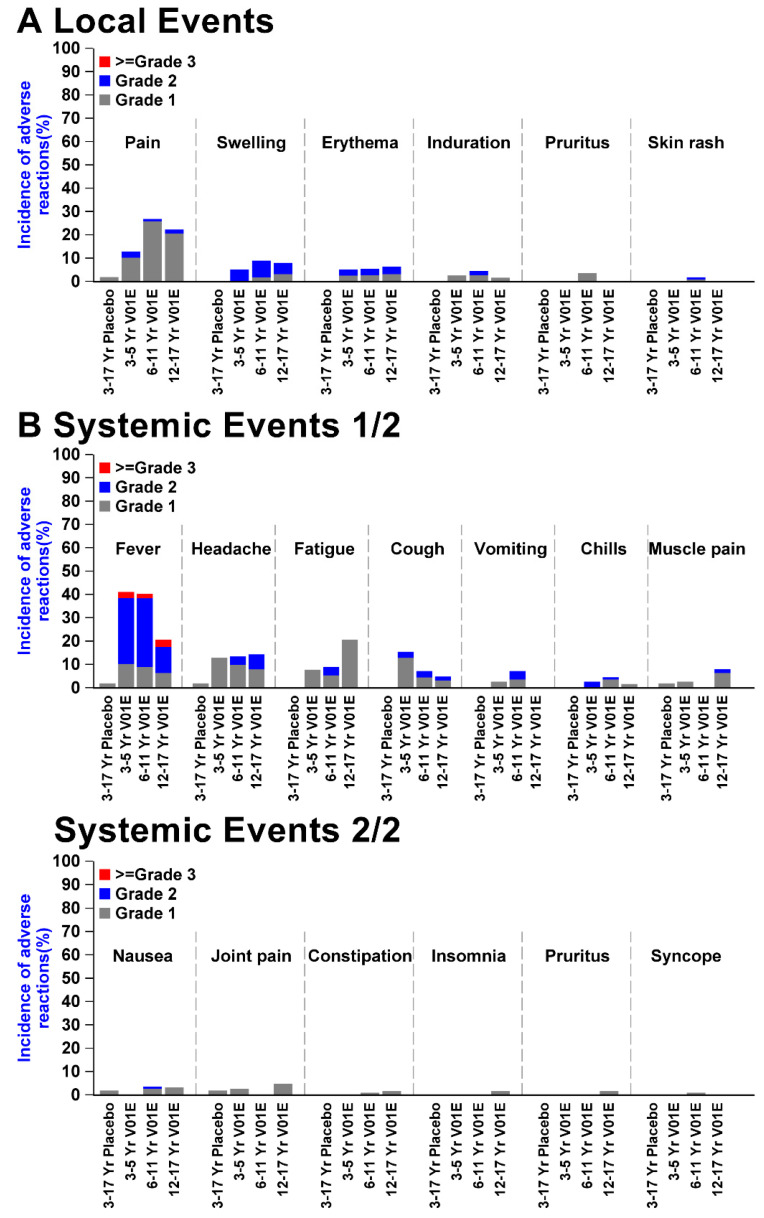
SCTV01E safety profile in 3 to 17 YOA population. Participants who received SCTV02E reported more injection site pain, swelling, and erythema than those who received a placebo (*p* < 0.001). The 12~17 years old participants reported less fever than the 3~11 years (*p* = 0.012), but more fatigue (*p* = 0.036). Other AE incidences were rare and similar between age groups.

**Figure 3 vaccines-13-00043-f003:**
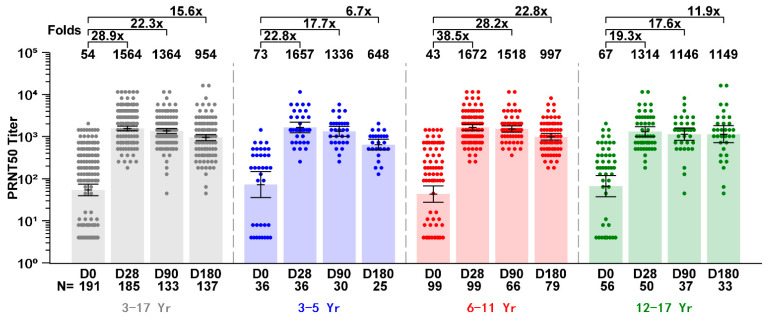
SCTV01E induced neutralizing antibody against SARS-CoV-2 Omicron BA.5 in 3 to 17 YOA population. Columns refer to GMT with values above and folds compared with baseline as well, error bars refer to 95% CI of GMT, scatterplots refer to neutralizing antibody titer of each participant. Neutralizing antibody titers in each age group were significantly boosted on Day 28, 90, and 180 compared with baseline (*p* < 0.001), with high fold increase.

**Table 1 vaccines-13-00043-t001:** Demographic characteristics of the participants aged 3 to 17 years and ≥18 years.

	3–17 YOA, n (%)	≥18 YOA (Phase 3 MRCT), n (%)
Sample size	214 (100)	188 (100)
Age at vaccination (years)		
Mean (SD)	9.1 (3.50)	52.4 (15.13)
Median (range)	8.0 (3–17)	53.5 (18–82)
Sex—n (%)		
Male	116 (54.2)	127 (67.6)
Female	98 (45.8)	61 (32.4)
History of COVID-19 vaccination		
Two-dose vaccination	214 (100.0)	10 (5.3)
Boosted vaccination	0	178 (94.7)
Types of last received COVID-19 vaccine		
Inactivated vaccine	214 (100.0)	121 (64.4)
Non-inactivated vaccine	0	67 (35.6)
Vaccination intervals		
6–12 months	6 (2.8)	79 (42.0)
12–24 months	208 (97.2)	109 (58.0)
BMI (kg/m^2^)		
Mean (SD)	17.1 (3.13)	24.0 (3.24)
Median (range)	16.2 (12.8–28.3)	23.5 (18.3–37.2)

**Table 2 vaccines-13-00043-t002:** SARS-CoV-2 serum neutralization assay results 30 days after SCTV01E injection among participants without evidence of infection.

Age Group	No. of Participants	Geometric Mean 50% Neutralizing Titer (95% CI)	Geometric Mean Ratio (95% CI), 3–17 Years vs. ≥18 Years	Difference Between the Percentage Achieved Seroresponse (95% CI), 3–17 Years vs. ≥ 18 Years
3–17 years	95	1290 (1077, 1545)	8.78 (6.05, 12.74)	6.34 (0.93, 11.22)
≥18 years	188	123 (98, 155)	—	—

## Data Availability

The sponsor, investigator, and other collaborators will review and approve data sharing proposals directed to the corresponding author based on scientific merit. Anonymized participant data will be made available and shared through a secure online platform after signing a data access agreement. More details including safety and immunogenicity are available in the Appendix A.

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
