# Peer review of "Safety and Immunogenicity of the Tetravalent Recombinant COVID-19 Protein Vaccine SCTV01E in Children and Adolescents Aged 3 to 17 Years: A Randomized, Double-Blind, Placebo-Controlled Phase 2 Clinical Trial"

_vaccines, 2025, doi:10.3390/vaccines13010043_

Round 1

Reviewer 1 Report

Comments and Suggestions for Authors

This manuscript describes a phase II study of a new multivalent vaccine in children. The goal was to monitor for adverse reactions and immunogenicity in children. The study was well presented and the results were clear. There are a couple of issues to address:

Figure 2: The graph is small and there is no indication of statistical significance. I recommend to change the order on x-axis showing youngest to oldest groups, and to break fig 1B into two graphs so that the figures with fonts can be enlarged to make it easier to read. There should also be indications of statistical differences, and if there is none, indicate this in figure legend.

Figure 3. The figure legend needs more details on what is being shown. The reader should be able to look at graph and read figure legend to understand everything in graph. There are titers along with how many fold difference from control group, but this is not clearly stated. The are grey bars, presumably showing means. There is a scatterplot along with error bars, which are not defined. In addition, there are no indications of statistical differences. The figure legend needs to provide these details and there needs to be some additional labeling in the figure.

Author Response

Comments 1: Figure 2: The graph is small and there is no indication of statistical significance. I recommend to change the order on x-axis showing youngest to oldest groups, and to break fig 1B into two graphs so that the figures with fonts can be enlarged to make it easier to read. There should also be indications of statistical differences, and if there is none, indicate this in figure legend.

Response 1: The order on x-axis was changed according to advices. And Fig 1B was broken into two graphs to make it easier to read. Statistical differences were added to the legend.

Comments 2: Figure 3. The figure legend needs more details on what is being shown. The reader should be able to look at graph and read figure legend to understand everything in graph. There are titers along with how many fold difference from control group, but this is not clearly stated. The are grey bars, presumably showing means. There is a scatterplot along with error bars, which are not defined. In addition, there are no indications of statistical differences. The figure legend needs to provide these details and there needs to be some additional labeling in the figure.

Response 2: More details were added to the legend, including fold difference from control group, definitions, and statistical differences.

Reviewer 2 Report

Comments and Suggestions for Authors

The paper describes a straightforward and routine vaccine test in children and adolescents aged 3 to 17 years. SCTV01E is a tetravalent recombinant COVID-19 protein vaccine authorized for use in China for adults 18 years and older but not for those under 18. The Phase 2 trial assessed the safety and immunogenicity of SCTV01E in healthy children and adolescents aged 3 to 17 years, to determine the immunogenicity in youths with that previously observed for adults from the original efficacy trials. Determinations were for safety and immunogenicity. No serious adverse effects of the injections were identified. Mild adverse effects were reported (i.e., soreness at the injection site, headache, chills or fever), but these adverse effects are common with COVID-19 vaccines and other vaccines.

The paper is well-written and describes the Phase 2 trial in detail.

This is a professionally prepared manuscript and study.

This reviewer did not find deficiencies in the acquisition or presentation of the data.   

Author Response

Comments: This reviewer did not find deficiencies in the acquisition or presentation of the data.

Response: Thanks for reviewing.

Reviewer 3 Report

Comments and Suggestions for Authors

This is a straight-forward safety and immunogenicity study of a Chinese made tetravalent protein vaccine against COVID-19. The vaccine contains trimeric spike proteins of Alpha, Beta, Delta and Omicron BA.1 variants. In adults the vaccine has shown about 70% efficacy against symptomatic COVID-19.

Against this background a phase I/II trial is a sheer formality. It turned out that the vaccine is also highly immunogenic in children, who had previously received two doses of whole cell COVID-19 vaccine. This is hardly surprising but it is difficult to assess the magnitude of immune responses. The Table 2 suggests that neutralizing antibody responses in 3–17-year-olds were superior to those in adults, but the Discussion says that the immune responses were only “non-inferior”. This should be clarified.

The safety results are reported in narrative and graphically in Fig.2. Fig.2 is not very informative, and the whole Fig. might be deleted. Alternatively, a smaller Fig. showing only fever and, possibly, pain, might be prepared to replace the current Fig.2.

Finally, the paper should contain some statement regarding the licensure and current use of the tetravalent vaccine in China.

Author Response

Comments 1: Against this background a phase I/II trial is a sheer formality. It turned out that the vaccine is also highly immunogenic in children, who had previously received two doses of whole cell COVID-19 vaccine. This is hardly surprising but it is difficult to assess the magnitude of immune responses. The Table 2 suggests that neutralizing antibody responses in 3–17-year-olds were superior to those in adults, but the Discussion says that the immune responses were only “non-inferior”. This should be clarified.

Response 1: The primary endpoint of the study was to prove that the immunogenicity of SCTV01E in 3–17-year-olds were non-inferior to that in adults (18 years and older) according to compare the neutralizing antibody mean titer ratio and the difference in seroresponse rate. The primary endpoint was met, further, the results were superior to those in adults. We would make a supplementary description about the superior results in the revised manuscript.

Comments 2: The safety results are reported in narrative and graphically in Fig.2. Fig.2 is not very informative, and the whole Fig. might be deleted. Alternatively, a smaller Fig. showing only fever and, possibly, pain, might be prepared to replace the current Fig.2.

Response 2: Figure 2 would be replaced with a smaller figure according to reviewer’s advice in which Figure 1B was broken into two graphs to make it easier to read. And more details would be added in the figure legend.

Comments 3: Finally, the paper should contain some statement regarding the licensure and current use of the tetravalent vaccine in China.

Response 3: The licensure and current use of the tetravalent vaccine in China would be added to the discussion section in the revised manuscript as follow: SCTV01E is applied in immunization as a booster dose in population aged 18 years and older in China according to Emergency Use Authorization, and the Biologics License Application process is ongoing.